# The Effect of Circle of Willis Morphology on Retinal Blood Flow in Patients with Carotid Stenosis Measured by Optical Coherence Tomography Angiography

**DOI:** 10.3390/jcm12165335

**Published:** 2023-08-16

**Authors:** Zsuzsanna Mihály, Lilla István, Cecilia Czakó, Fruzsina Benyó, Sarolta Borzsák, Andrea Varga, Rita Magyar-Stang, Péter Vince Banga, Ágnes Élő, Róbert Debreczeni, Illés Kovács, Péter Sótonyi

**Affiliations:** 1Department of Vascular and Endovascular Surgery Heart and Vascular Centre, Semmelweis University, 1122 Budapest, Hungary; zsuzsannamihaly@gmail.com (Z.M.);; 2Department of Ophthalmology, Semmelweis University, 1085 Budapest, Hungaryelo.agnes@med.semmelweis-univ.hu (Á.É.); 3Department of Neurology, Semmelweis University, 1085 Budapest, Hungarydebreczeni.robert@med.semmelweisuniv.hu (R.D.); 4Department of Ophthalmology, Weill Cornell Medical College, New York, NY 10021, USA; 5Department of Clinical Ophthalmology, Faculty of Health Sciences, Semmelweis University, 1085 Budapest, Hungary

**Keywords:** carotid artery stenosis, Circle of Willis, retinal blood flow, cerebral microcirculation

## Abstract

The Circle of Willis (CoW) is the main collateral system, and its morphological variants are more common in patients who have severe carotid artery stenosis. Earlier data suggest that optical coherence tomography angiography (OCTA) may help to assess the changes in cerebral vascular perfusion by imaging the retinal blood flow. In this single-center prospective clinical study, patients scheduled for carotid endarterectomy (CEA) underwent preoperative computed tomography angiography (CTA) of the extra- and intracranial cerebral circulation. OCTA imaging was performed one week before surgery and postoperatively one month later. The patients were divided into two subgroups based on CTA evaluation of CoW: compromised CoW or non-compromised CoW (containing hypoplastic and normal segments). The effect of the patient’s age, OCTA scan quality (SQ), CoW morphology, laterality, and surgery on superficial capillary vessel density (VD) in the macula were assessed in multivariable regression models using linear mixed models. We found that VD significantly decreased with aging (−0.12%; 95%CI: −0.07–−0.15; *p* < 0.001) and was significantly higher in patients with non-compromised CoW morphology (by 0.87% 95%CI (0.26–1.50); *p* = 0.005). After CEA, retinal blood flow significantly improved by 0.71% (95%CI: 0.18–1.25; *p* = 0.01). These results suggest that in the case of carotid artery occlusion, patients with non-compromised CoW have more preserved ocular blood flow than subjects with compromised CoW due to remodeling of the intra-orbital blood flow. Measuring the retinal blood flow might be used as a relevant and sensitive indicator of collateral cerebrovascular circulation.

## 1. Introduction

Prevention of cerebral ischemia is the goal of pharmacological or surgical treatment of carotid artery stenosis [1,2]. In the case of asymptomatic patients with carotid stenosis, the number of patients needed to treat to reach one preventable stroke event with endarterectomy is 20, which is high compared to 6 in the symptomatic patient group [3,4]. There is a clear need to select asymptomatic patients on the best medical treatment regime at high risk of ischemic events to avert unnecessary surgery and with more benefits, such as better intracranial blood flow after carotid reconstruction. 

The current guideline for cerebrovascular disease recommends carotid endarterectomy as the best medical therapy for asymptomatic significant carotid artery stenosis patients with low surgical risk [1,2]. A current meta-analysis assessing the relevance and feasibility of risk/benefit-oriented selection of asymptomatic patients for revascularization concluded that extension of routine assessment (such as cerebrovascular reserve capacity measurement by transcranial doppler (TCD), plaque progression, and morphology by ultrasound or MR) of asymptomatic stenosis beyond the grade of stenosis may help to improve the risk management and optimize therapy [5]. The ophthalmic symptoms (even if it is permanent loss of vision) and their imaging are underrepresented in the decision-making process of symptomatic and asymptomatic carotid artery reconstructions. 

### 1.1. Retinal Blood Flow Measurement in Patients with Carotid Artery Stenosis

The retina is vulnerable to ischemia since it is a high-energy-consuming tissue in the eye. The retinal microcirculation and structure can be viewed directly, offering an accessible route to monitor vascular and circulatory function. As a part of the central nervous system, the retina offers a unique and obvious way to study cerebral small vessel diseases in clinical routine [6]. In the last decades, numerous novel diagnostic tools in retinal imaging have been developed that may allow researchers and physicians to understand the genesis and progression of cerebrovascular diseases. Reversed flow direction in ophthalmic artery detected by color flow duplex ultrasonography is highly specific for severe ipsilateral ICA stenosis or occlusion, with high positive predictive value, moderate negative predictive value, and limited sensitivity. Furthermore, it may also give additional information on the hemodynamic function and the inadequacy of other collateral vessels, such as the anterior communicating artery [7]. 

Optical coherence tomography angiography (OCTA) is a relatively new approach for visualizing and analyzing retinal and choroidal vasculature without intravenous dye. With the help of OCTA, subtle changes in retinal blood flow can be detected with high accuracy. It utilizes motion contrast technology to detect red blood cell movement within vessels. This allows for accurate visualization of the microvasculature in the macular area and around the optic disc head, providing valuable quantitative information like vessel density and foveal avascular zone size. The procedure is fast, repeatable, and offers both structural and blood flow data simultaneously. Since it is a fast and non-invasive examination, the advantage of OCTA is that it is comfortable for patients and easy to repeat at any time during follow-up visits. The novel OCTA devices are able to distinguish the superficial and the deep capillary plexus of the retina. The superficial retinal capillary plexus, which is supplied by the central retinal artery, is evaluated during the measurements. Several studies give evidence of the high reproducibility and accuracy of OCTA parameters in control groups [8,9,10,11,12,13,14,15], as well as in patients with diabetes mellitus, glaucoma [16], ischemic optic neuropathy [17], or retinal vascular diseases [18,19]. The use of OCTA gives the opportunity of detecting early signs of retinal microvascular abnormalities, even minor changes in the intracranial blood flow [6]. The ophthalmological imaging features of OCTA have been studied in patients with carotid artery stenosis and were found to be more sensitive than color flow duplex ultrasonography. Reduced retinal flow density compared to healthy subjects and a significant improvement in the capillary network were described after carotid reconstruction [20,21]. Furthermore, a significant increase in the vessel density of the macular deep vessel complex in both eyes after surgical procedures was also reported [20,21]. 

### 1.2. Role of CoW and Cerebral Collateral Blood Flow in Patients with Carotid Stenosis

The most important collateral pathway to maintain the perfusion of the affected vascular territory is the CoW [22,23]. The anatomical characteristics of the CoW play an important role in the outcome of carotid artery stenosis. The anterior (ACoA) and posterior communicating arteries (PCoA) are considered the primary collateral pathways in patients with internal carotid artery (ICA) stenosis. Preferential flow patterns may develop, such as the presence of collateral flow via the ACoA, the PCoA, or both. Previous studies analyzed the CTA scans of patients who underwent carotid artery reconstruction and found that in patients with significant carotid artery stenosis, the prevalence of CoW variants was significantly higher compared to a healthy control group [24,25]. It was also noted that absent or hypoplastic segments were more common among stroke patients [24,25]. Other collateral vessels (leptomeningeal collaterals and extracranial to intracranial collaterals, for example, with retrograde flow in the ophthalmic artery) are acknowledged as secondary cerebral collateral blood flow. There are further cortical anastomoses extending between the terminal branches of the cerebral arteries at the surface of the cortex and into the leptomeningeal space, just before the pial arteries penetrate the gray matter. There is also a collateral supply system between the external carotid artery and the ICA. 

The purpose of our single-center prospective study (NCT03840265) was the quantitative analysis of the retinal microvascular perfusion on both eyes before and after carotid artery endarterectomy in relation to compromised or non-compromised CoW morphology. 

## 2. Materials and Methods

### 2.1. Patients’ Enrollment and Surgical Treatment

The clinical data and outcomes of all consecutive patients who were enrolled after signing the informed consent to the study considering the exclusion and inclusion criteria of the study (Table 1), were collected. 

Hypertension was defined according to the 2020 American Heart Association guideline [26]. Diabetes was defined in line with the 2019 European Society of Cardiology guideline [27]. Chronic kidney disease (CKD) was defined according to the kidney disease: Improving Global Outcomes (KDIGO) [28]. Chronic obstructive pulmonary disease was defined according to Global Initiative for Chronic Obstructive Lung Disease consensus report [29]. The best medical therapy (Aspirin 100 mg once or Clopidogrel 75 mg once with a statin) was prescribed for all patients after the result of the CT angiography according to the European Society for Vascular Surgery guidelines [2]. Five experienced vascular surgeons performed surgical carotid artery reconstructions under general anesthesia. The majority of patients received eversion endarterectomy, with selective shunting and bovine patch angioplasty closure. After the procedure, patients underwent a basic neurological evaluation and were discharged on the third postoperative day. Single antiplatelet therapy was prescribed, except for cases where longer dual antiplatelet therapy was required due to cardiac indications.

All patients underwent pre- and postoperative OCT examination in Semmelweis University, Department of Ophthalmology and CEA (carotid artery endarterectomy) at Semmelweis University, Department of Vascular and Endovascular Surgery, between 1 January 2019 and 31 January 2021. 

The study was approved by the institutional ethical committee (SE-KREB 84/2019) (IV/667-1/2022/EKU) and was carried out in accordance with the Declaration of Helsinki.

### 2.2. Brain CT and CTA Examinations and Evaluation

All CT imaging scans were performed on a 256-slice scanner (Brilliance iCT 256, Philips Healthcare, Best, The Netherlands). Brain CT was used (according to the institutional, local protocol) following the parameters: field of view 200–250 mm, collimation 64 × 0.625, pitch 0.39, gantry rotation time 400 ms, tube voltage 120 kVp, tube current 120–204 mAs, slice thickness 2 mm, dose-length product 312–626 mGycm. Contiguous reconstruction was performed with 0.67-mm slice thickness and 512 × 512 matrix using iterative model reconstruction (IMR, Philips Healthcare, Cleveland, OH, USA). Thin section images were thoroughly evaluated on an IntelliSpace Portal workstation (Philips Healthcare, Best, The Netherlands), including 3 mm maximum intensity projection (MIP) slabs parallel and perpendicular to the anterior skull base to provide the best overview of the CoW. 

On both sides, ICA stenosis was determined according to the NASCET method [30]. The ipsilateral ICA stenosis was divided into two subgroups 70–89% and 90–99% stenosis [31]. The CoW was evaluated according to the evaluation routine described before [24]. Each individual segment was scored as normal (diameter ≥ 0.8 mm), hypoplastic (<0.8 mm), or non-visualized. Groups were formed as follows: (I) compromised CoW contains at least one non-visualized segment; (II) non-compromised CoW contains hypoplastic and normal segments. Figure 1 displays some examples of the CoW subgroups. 

### 2.3. OCTA Examination

Each study subject underwent three sessions of imaging, during which three OCTA images of the macular area were obtained consecutively. The first measurement was scheduled during the preoperative period; the other two examinations were performed during the first postoperative week and one month following the surgery. Ophthalmologic examinations consisted of testing visual acuity, slit lamp, fundus examinations, along with OCTA imaging. One trained examiner performed all the ophthalmological imaging. OCTA imaging was performed using the AngioVue device with an SSADA (split-spectrum amplitude-decorrelation angiography) software algorithm (RTVue XR Avanti with AngioVue, OptovueInc, Fremont, CA, USA). The segmentation of the obtained scans and the quantitative analysis are performed by the built-in software. The device obtains 70,000 A-scans/sec in appr. 3.0 s. The required imaging of the macula needs a 3 × 3 mm scan by the current version of AngioAnalytics software version 2017.1, phase 7 update (acquired scans with the highest resolution in the central 3 mm diameter). Images with movement- (such as vessel doubling, white line artifacts, vessel discontinuities, or noise) and projection artifacts and segmentation errors were excluded if they appeared in the imaging process. Scans with motion or blink artifacts were also excluded. Scan quality (SQ) above 5 was required for inclusion. Angiograms with a signal quality of 5 or lower were excluded from the study; only those above 5 were used.

### 2.4. Statistical Analysis

The statistical analysis was performed with SPSS software (version 23.0, IBM, Armonk, NY, USA). The effect of CoW morphology (isolated or non-visualized versus normal or hypoplastic segments), carotid endarterectomy, laterality, and age on macular vessel density were assessed with multivariable regression analysis with general estimating equation (GEE) models. In addition to treating repeated measurements (intrasession and between visits), this test enables adjustments to be made for within-subject correlation of parameters (contralateral versus ipsilateral eye) by considering between-eye correlations. Furthermore, the inclusion of scan quality and not just other covariates into GEE models allows one to simultaneously control for their effect on the dependent variables as well.

## 3. Results

### 3.1. Enrollment

Between 2 February 2019 and 30 July 2020, 56 enrolled patients underwent preoperative CTA of the extra- and intracranial cerebral circulation and pre- and postoperative OCT in our prospective study. In the 36 male and 20 female patients, the mean age was 69.89 ± 7.07 years (non-compromised CoW 67.00 ± 6.60; compromised CoW 70.35 ± 6.92, *p* = 0.18), and the mean degree of stenosis was 79.42 ± 8.90 (non-compromised CoW 80.38 ± 5.94; compromised CoW 79.14 ± 9.66 *p* = 0.62). There was no difference in the incidence of near occlusion (90–99% degree of stenosis) and significant stenosis (70–89% degree of stenosis), additional data has been added in the Appendix A Table A1. Table 2 summarizes patients’ demographics, comorbidities, and medical therapy. 

### 3.2. CoW Analysis Based on CT Angiography

Patients were divided into compromised and non-compromised CoW groups. Table 3 shows the incidence of hypoplastic, normal, and non-visualized segments of patients’ CoW morphologies. Patients with ipsilateral 70–89% stenosis and subocclusive (90–99%) stenosis had no significant difference in the incidence of CoW morphology (data provided in the Appendix A). Only in the case of aspirin intake was there a significant difference in clinical characteristics between these subgroups. 

### 3.3. OCT Measurements

SQ values of the OCTA measurements on 112 eyes of 56 patients ranged from 6 to 10, the overall mean SQ was 7.55 ± 0.99, and the mean superficial macular vessel density was 42.96 ± 4.13%. Since scan quality is a well-known significant predictor of superficial capillary vessel density in the macula, SQ values associated with each measurement were included as confounders in statistical calculations.

The multivariable analysis, after adjusting for SQ values, identified the following parameters as significant predictors of retinal vessel density: age, CoW morphology, and effect of carotid reconstruction (Table 4). We found no statistically significant difference between ipsi- and contralateral vessel densities (Table 4).

## 4. Discussion

According to our findings, ocular blood flow is higher in patients with non-compromised CoW than in subjects with compromised CoW. The significantly increased perfusion of retinal capillaries in both eyes after CEA suggests that OCTA can detect the improvement in ocular blood flow in these patients and might help to give insight into cerebral microcirculatory changes after CEA. These results suggest that the assessment of ophthalmic blood flow might be used as a relevant and sensitive indicator of collateral cerebrovascular circulation.

### 4.1. CoW Categorization

CoW structure is supplied by afferent arteries from the two ICAs and the basilary artery. In imaging studies, complete morphology of the entire CoW was reported in 27–90% of healthy individuals and 18–55% in cerebrovascular diseases [24,32]. The incidence of complete CoW is higher in women for all age groups compared to men, but it rises with age in both sexes [33]. The incidence of CoW variation in our patient group has not altered from the previously published data. There was no difference in sex distribution in our cohort, most likely because of the lower sample size. The incidence of CoW variation in the 70–89 and 90–99 subocclusive groups showed no difference.

Comparing the incidence of CoW variation with CTA studies on cerebrovascular patients, the prevalence of non-visualized AComA 3.6% A1 (3.6%) and the most compromised PcomA (35.7%) in our analysis was a bit less frequent (versus 4–15% and 41–66%, respectively) [24]. The incidence of compromised and non-compromised CoW showed no difference between the patients with 70–90% and 90–99% ipsilateral ICA stenosis. The collateral capacity of a vessel seems to be determined by luminal caliber based on the results of a TCD study. The threshold diameter allowing for cross-flow through the main collateral arteries was reported between 0.4 and 0.6 mm in the CoW in the above-mentioned study with TCD and postmortem CoW morphology analysis [34]. The normal and hypoplastic segments were defined with a threshold of 0.8 mm based on a previous publication on imaging the CoW [24]. The number of enrolled patients and the variable occurrence of the different segment variations do not allow further subgroup analysis involving the separation of hypoplastic and normal segments of CoW.

A study [30] including 38 patients with unilateral ICA occlusion showed that the pattern of collateral supply has a significant influence on the hemodynamic status. According to their results, collateral flow via the ACoA is a sign of a preserved hemodynamic status, whereas no collateral flow via the CoW or flow via only the PCoA is a sign of a deteriorated cerebral hemodynamic perfusion. Best-preserved hemodynamics was reported if both the anterior and posterior circles of the CoW were complete [35]. In our study, the number of compromised anterior circles of the CoW was low (*n* = 6) for further statistical analysis, so this hypothesis cannot be tested in our cohort. Furthermore, data from 11 enrolled patients with significant contralateral carotid artery stenosis or occlusion were also inadequate for statistical analysis. 

Previously, it was reported that carotid endarterectomy resulted in an increase in cerebral blood flow only if the degree of stenosis was greater than 90% [36]. There was no difference between the CoW compromised and non-compromised morphology in the subgroup of patients with 70–90 stenosis and 90–99% subocclusive stenosis in our study.

### 4.2. Role of Collateral Flow

Collateral circulation represents an important aspect of cerebrovascular disease, but it has remained largely underappreciated. ICA is the most important vessel in cerebral blood flow compensation, but the accentuated role of the external carotid artery and vertebral artery highlights their impact [37]. Arterial flow reduction due to extracranial or intracranial carotid artery stenotic disease promotes collateral recruitment in chronic cerebral hypoperfusion [38], although the relationship of these collaterals with cerebral blood flow and clinical symptomatology still remains unclear. The reinforcement from the collaterals likely depends on several compensatory hemodynamic, metabolic, and neural mechanisms, and its changes to cerebrovascular intervention are unknown. Diagnostic imaging of collaterals is limited to TCD, CTA, MR angiography, or conventional angiography [38]. OCTA can be a more sensitive and clinically easier applicable diagnostic tool to measure ophthalmic collateral flow. 

The collateral recruitment in the CoW and the clinical manifestations of carotid artery disease are highly variable. Simply because the anatomic variations in CoW and the contralateral or intracranial carotid and vertebral artery stenosis or occlusion can combine, the possible number of variations is quite high. Contralateral ICA mostly secures the bilateral anterior cerebral artery territory and not only the middle cerebral artery on the ipsilateral side to the stenosis [39]. Because of collateral recruitment, a compromised blood flow rate in the middle cerebral artery is not necessarily related to the degree of carotid stenosis [39]. The protective role of collaterals was found to be similar for both ACoA and PCoA in a study comparing blood flow via CoW by TCD in asymptomatic and symptomatic patients with ICA stenosis [40]. In our study, the effect of the patient’s age, scan quality of OCTA measurement, CoW morphology based on CTA imaging, stenosis laterality, and surgery on superficial capillary vessel density in the macula were assessed in multivariable regression models.

### 4.3. OCT Image Quality

Although image quality is known to significantly affect both measurement error [41,42,43,44,45] and OCTA parameters [46], it should be used as a correction factor for longitudinal analysis. Optical coherence tomography helps in visualizing the retinal structures with the use of low-coherence interferometry [47]. Media opacities, ocular saccades, blink artifacts, and OCT operator skills can affect OCTA scan quality. SQ index is a unitless parameter in the range from 1 to 10 produced by the RTVue-XR AngioVue software. Errors originate from eye motion, defocus, and signal-to-noise ratio, which are incorporated in SQ formulation. Image quality needs to be taken into consideration during the comparison of OCTA images for reliable results. Although OCTA can detect the ophthalmological presentation of carotid stenosis and might help to assess cerebral microcirculatory changes in these patients [48], its importance and potential to select patients for revascularization has not been fully recognized yet.

### 4.4. The Role of Ophthalmic Artery in Collateral Flow

Reversal of blood flow within the ophthalmic artery can give secondary collateral aid in patients with carotid artery stenosis. The orbital plexus connects the ophthalmic artery with facial, middle meningeal, maxillary, and ethmoidal arteries; thus, they can participate in secondary collateral flow in the case of chronic cerebral hypoperfusion. Patients with reversed ophthalmic artery flow in a previous case-control study showed a steal phenomenon, characterized by a shunt to the low-resistance intracranial circulation and a reduction in the retrobulbar blood flow [49]. Costa et al. have classified retrobulbar circulation into four types. According to this research group’s results, the effect of carotid revascularization on ocular hemodynamics was found to improve the ipsilateral retrobulbar blood flow among 17 patients [50]. In our results, improved retinal blood flow on both sides suggests superiority in cerebral collateral blood flow through the CoW.

### 4.5. Effect of CoW Morphology on Retinal Blood Flow 

The retina’s blood supply originates from the retinal and choroidal vasculature, both derived from the ophthalmic artery, the initial branch of the internal carotid artery. Analogous to cerebral microcirculation, the retinal system lacks anastomoses, serves as a barrier with autoregulatory capabilities, and functions as a relatively low-flow, high-oxygen-extraction system [51]. There are a few publications of animal models on retinal ischemic injury by uni- and bilateral common carotid occlusion with proof of retinal ischemia by blocking the collateral flow from CoW [52,53]. Previous intraoperative measurements suggested that orbital vessel monitoring can confirm the patency of the Circle of Willis. These findings suggested that the retinal blood flow can be “an acoustic window” into intracranial blood flow during cardiac surgery [54]. In the case of our study, patients with non-compromised CoW have better ocular blood flow than subjects with compromised CoW. The significant improvement in retinal capillary perfusion in both eyes after CEA suggests that there is a remodeling in intra-orbital blood flow after a revascularization procedure. There are no publications on the association of OCTA measurement with CoW anatomy in patients undergoing carotid endarterectomy. 

### 4.6. Positive Clinical Effect of Increased Retinal and Cerebral Blood Flow

As a third level of prevention, carotid artery reconstruction is the main and strongest indication for a patient after a neurological event. The recommendation at IIa level for asymptomatic patients with significant carotid artery stenosis in clinical guidelines [2] suggests open carotid reconstruction if cerebrovascular reserve capacity measurement by TCD, plaque progression, and morphology by ultrasound or MR [5] proves a higher risk for stroke. Currently, reconstruction is suggested as secondary stroke prevention for asymptomatic patients with high stroke risk. Thus, restoration of blood flow to the brain can alleviate ischemic symptoms and prevent further brain damage [55]. 

Currently, there is no consensus on the indication of carotid reconstruction to improve cerebral blood flow, although there are published results that indicate that the increased retinal blood flow after CEA is associated with improved visual acuity and preservation of visual field due to decreased retinal ischemia [20,21]. Furthermore, improved cerebral blood flow following carotid artery reconstruction has been linked to improved cognitive function [55], suggesting that reconstruction surgery might also be indicated in asymptomatic patients. Simultaneously decreased cerebrovascular reactivity and dysfunction of the cardiovascular autonomic nervous system in patients with severe atherosclerosis have important clinical implications in subsequent ischemic stroke risk estimation and personalized hemodynamic characterization [56]. Our results strongly support that in the case of carotid artery stenosis, patients with non-compromised CoW have more preserved ocular blood flow than subjects with compromised CoW due to remodeling of the intra-orbital blood flow. One might conclude from these results that patients with non-compromised CoW have more potential benefits due to improved retinal and possibly cerebral function, which can be an additional factor for carotid reconstruction indications. 

### 4.7. Limitations

Some limitations of our study must be mentioned. Sample size does not allow us to calculate the correlation of the isolated CoW and every variation of the CoW (most of all compromised anterior CoW) with the change of vessel density. One of the limitations is that the acquired data of the retinal vasculature were obtained using a specific type of OCTA device, impeding the generalizability of our results. A further limitation is the lack of longer follow-up and repeated OCT measurements or a control group with patients who did not undergo any revascularization or a control group with healthy individuals. Complementary diagnostic imaging examinations of CoW function (such as cerebral MR, TCD, and diagnostic cerebrovascular angiography) were not performed in the full cohort of patients. There are no available invasive intraoperative measurements of the backflow or imaging to estimate other possible hemodynamic compensation mechanisms (leptomeningeal collateral flow) besides CoW and ocular collateral flow in all enrolled patients. 

## 5. Conclusions

In conclusion, it can be stated that in the case of patients with non-compromised CoW, who are candidates for carotid reconstruction, ocular blood flow is better than in subjects with compromised CoW. The use of OCTA, a non-invasive and easily repeatable new imaging modality, provides an opportunity to detect early signs of retinal microvascular abnormalities, which might mimic and mirror changes in the intracranial blood flow. The significant improvement in retinal capillary perfusion in both eyes after CEA suggests that there is a remodeling in intraorbital blood flow after a revascularization procedure, and OCTA is able to detect the ophthalmological presentation of carotid stenosis and might help to assess cerebral microcirculatory changes in patients with carotid artery stenosis. OCTA measurement combined with CoW morphology can offer a relevant aspect in achieving a benefit-oriented selection of patients for revascularization procedures.

## Figures and Tables

**Figure 1 jcm-12-05335-f001:**
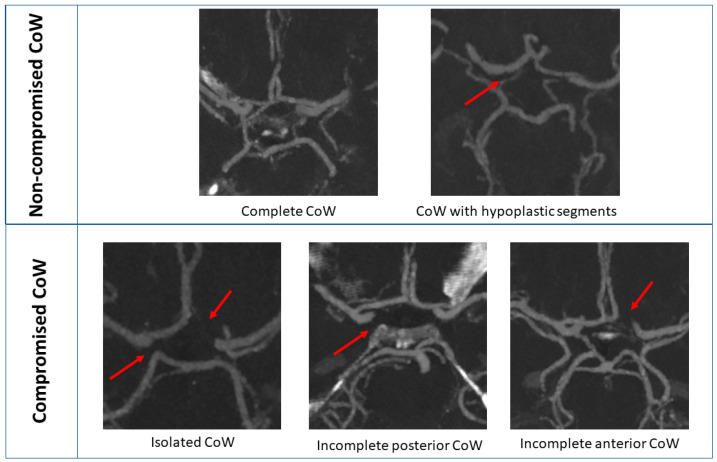
Maximum intensity projection of CTA provides an overview of compromised and non-compromised CoW. The red arrow shows the hypoplastic or non-visualized segments. CTA: CT angiography, CoW: Circle of Willis.

**Table 1 jcm-12-05335-t001:** Inclusion and exclusion criteria of the study.

Inclusion Criteria	Exclusion Criteria
Signed the informed consent	Withdrawal of the informed consent
*Criteria of the vascular surgery part*	
Significant carotid artery stenosis (≥70%) NASCET criteria in carotid CTA	Pacemaker implantation
Planned endarterectomy	Chronic kidney disease in Stage V
Age under 50 years	
Neurological event 15 days before operation	
*Criteria of the carotid CT angiography*	
Examination by Brilliance iCT 256	Serious movement artifacts
*Criteria of the OCTA examination*	
	Associated ocular disease (age-related macular degeneration, glaucoma, vitreomacular disease)
	Previous intraocular anti-VEGF injection
	Clinically significant media opacities Serious movement artifacts or scan quality below 5

CTA: CT angiography, OCTA: optical coherence tomography angiography, NASCET: North American Symptomatic Carotid Endarterectomy Trial, VEGF: vascular endothelial growth factor.

**Table 2 jcm-12-05335-t002:** Clinical characteristics of study subjects and Circle of Willis subgroups.

	All Patients	Non-Compromised CoW	Compromised CoW	*p*-Value
Number of patients n (%)	56	13	43	
Sex female n (%)	17 (30.4%)	4 (9.3%)	16 (37.2%)	0.75
Contralateral ICA significant stenosis/occlusion n (%)	10 (17.9%)	4 (9.3%)	6 (13.9%)	0.19
Ipsilateral ICA stenosis 70–89%	44 (78.6%)	11 (84.6%)	33 (76.4%)	0.71
Ipsilateral ICA stenosis 90–99%	12 (21.4%)	2 (15.4%)	10 (23.25%)	
Symptomatic ICA stenosis n (%)	4 (7.1%)	1 (7.7%)	3 (6.9%)	1.00
Comorbidities				
Smoking n (%)	16 (28.6%)	6 (46.1%)	10 (23.3%)	0.16
Hypertension n (%)	52 (92.9%)	12 (92.3%)	40 (93.0%)	1.00
Diabetes mellitus n (%)	21 (37.5%)	3 (23.1%)	18 (41.9%)	0.33
Ischemic heart disease n (%)	13 (23.2%)	6 (46.1%)	7 (16.3%)	0.05
COPD n (%)	5 (8.9%)	2 (15.4%)	3 (6.9%)	0.58
Medical therapy				
Aspirin n (%)	33 (58.9%)	9 (69.2%)	24 (55.8%)	0.52
Clopidogrel n (%)	21(37.5%)	7 (53.8%)	14 (32.6%)	0.20
Statin n (%)	33 (58.9%)	7 (53.8%)	26 (60.5%)	0.75

n: number, ICA: internal carotid artery, COPD: Chronic obstructive pulmonary disease.

**Table 3 jcm-12-05335-t003:** Incidence of normal, hypoplastic, and non-visualized segments of CoW morphologies.

All (n = 56)	Normal Segment	Hypoplastic Segment	Non-Visualized Segment
AcomA (n)	94.6% (53)	1.7% (1)	3.6% (2)
A1 (n × 2)	91.9% (103)	4.5% (5)	3.6% (4)
PcomA (n × 2)	44.6% (25)	19.6% (11)	35.7% (20)
P1 (n × 2)	86.6% (97)	8.0% (9)	5.4% (6)
Anterior CoW semicircle	64.3% (36)	25.0% (14)	10.7% (6)
Posterior CoW semicircle	37.5% (21)	28.6% (16)	37.5% (21)

AcomA: anterior communicating artery, A1: anterior cerebral artery, PcomA: posterior communicating artery, P1: posterior cerebral artery, CoW: circle of Willis.

**Table 4 jcm-12-05335-t004:** Predictors of superficial macular vessel density after controlling for the effect of scan quality.

Predictors	ß	Confidence Interval	*p*-Value
Scan quality	1.80	1.53–2.07	<0.001
Age * (years)	−0.12	−0.07–−0.15	<0.001
CoW morphology * (non-compromised vs. compromised)	0.87	0.26–1.50	0.005
Carotid endarterectomy *	0.71	0.18–1.25	0.01
Laterality * (ipsi- vs. contralateral to the reconstruction)	0.39	−0.13–0.91	0.14

Note: * bivariable regression analysis after controlling for the effect of scan quality.

## Data Availability

Our prospective study has been registered under ClinicalTrials.gov Identifier: NCT03840265, further information about the study can be found there.

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
