# Peer review of "The Effect of Circle of Willis Morphology on Retinal Blood Flow in Patients with Carotid Stenosis Measured by Optical Coherence Tomography Angiography"

_jcm, 2023, doi:10.3390/jcm12165335_

Round 1

Reviewer 1 Report

Mihaly et al aimed to quantitatively analyze the retinal microvascular perfusion on both eyes before and after carotid artery endarterectomy in relation to compromised or non-compromised CoW morphology. The work is interesting and novel, however, I have some suggestion:

Line 68, the first use of OCTA should be spelled out in full. Plus not all readers would understand OCTA so I suggest the author give a well-detailed information on the OCTA and also previous studies that have used this tool.

There is little information on the characteristics of patients enrolled in the study and the authors have to give a detailed information on the patients used.

Line 145. With the specification of the OCTA, the authors did not make mention of which retinal plexus was used in their project and also should define the plexus that was used

Also, how were the angiograms assessed? Plus with a very low signal quality of 5, angiograms would be difficult to assess.

it would be interesting to compare the CoW structure in patients and controls

was there an association between the CoW and OCTA parameters? could the authors explain it more in the discussion section?

the discussion part is quite interesting but I suggest the authors discuss more about the effect of Circle of Willis morphology on retinal blood flow in patients with carotid stenosis but showing their association

Refer to this manuscript: PMID: 35892420

Author Response

Thank you for reviewing our study. The manuscript is revised and uploaded. Our responses for your questions and comments are given in more detail below.

  1. Line 68, the first use of OCTA should be spelled out in full. Plus not all readers would understand OCTA so I suggest the author give a well-detailed information on the OCTA and also previous studies that have used this tool.
    • OCTA abbreviation will be released in the text once more. We have added more detail to the introduction section about OCTA.
  2. There is little information on the characteristics of patients enrolled in the study and the authors have to give a detailed information on the patients used.
    • The definition of the different patient characteristics have been added to the methods section. The brief description of the procedure has been added as well to the methods section. More patient characteristics are added to the text and the manuscript including the degree of stenosis in the results section.  
  3. Line 145. With the specification of the OCTA, the authors did not make mention of which retinal plexus was used in their project and also should define the plexus that was used Also, how were the angiograms assessed? Plus with a very low signal quality of 5, angiograms would be difficult to assess.

    • The OCTA device used is able to distinguish the superficial and the deep capillary plexus of the retina. In this study we evaluated the superficial retinal capillary plexus, which is supplied by the central retinal artery. The segmentation of the obtained scans and the quantitative analysis is performed by the built in software. Angiograms with a signal quality of 5 or lower were excluded from the study, only the ones above 5 were used. Scans with motion or blink artefacts were also excluded. This information has been highlighted in Table 1 and in the methods section in the manuscript.
  4. it would be interesting to compare the CoW structure in patients and controlswas there an association between the CoW and OCTA parameters? could the authors explain it more in the discussion section?
    • In our prospective study design, there was no control group of patients without carotid artery stenosis or patients with carotid artery stenosis, who did not underwent reconstruction. This has been added to the limitation section.
  5. the discussion part is quite interesting but I suggest the authors discuss more about the effect of Circle of Willis morphology on retinal blood flow in patients with carotid stenosis but showing their association
    • A new paragraph with the subheading "Effect of CoW morphology on retinal blood flow" has been added to the discussion section to provide more insight to this topic.

Thank you for your review, we hope our answers here and modifications in the manuscript will provide exact information for you and fulfill the major revision which was asked from you.

Reviewer 2 Report

I read with interest the article by Zsuzsanna Mihály et al. regarding the quantitative analysis of the retinal microvascular perfusion on both eyes before and after carotid 104 artery endarterectomy in relation to compromised or non-compromised CoW morphology.

The authors suggest that in cases of carotid artery occlusion, patients with non-compromised CoW have more preserved ocular blood flow than subjects with compromised CoW due to remodeling of the intraorbital blood flow.

The manuscript is well written and of interest, but I have some recommendations to increase the quality:

1. In Table 2, I suggest the authors present carotid stenosis in more detail. As they mentioned in the materials and methods section, they enrolled patients with >70% stenosis. I suggest the authors to present the stenosis according to the degrees of severity, such as 70-90% 90-99% (subocclusive), respectively occlusion. Please see the following article: https://doi.org/10.3390/ijerph192113934

2. Additionally, it would be interesting if all the variables were presented depending on the severity of the stenosis (occlusion vs non-occlusion)

3. In Table 3, I suggest the authors to present the statistical analysis and the statistically significant difference between the 3 groups.

Author Response

Thank you for reviewing our study. The manuscript is revised and uploaded. Our responses for your questions and comments are given in more detail below.

  1. In Table 2, I suggest the authors present carotid stenosis in more detail. As they mentioned in the materials and methods section, they enrolled patients with >70% stenosis. I suggest the authors to present the stenosis according to the degrees of severity, such as 70-90% 90-99% (subocclusive), respectively occlusion. Please see the following article: https://doi.org/10.3390/ijerph192113934
    • The methods section is supplemented with disease definitions. We have added the 70-90% and 90-99% stenosis degree subgroups to the Table2 in the results section and a paragraph in the discussion section is added as well.
  2. .  Additionally, it would be interesting if all the variables were presented depending on the severity of the stenosis (occlusion vs non-occlusion):
    • We have added the 70-90% and 90-99% stenosis degree subgroups to Table2. If there is a need for it we provide an extra Table 5 as supplement with the baseline characteristics and CoW status of the 70-90% and 90-99% subgroup, but further subgroup analysis was not possible due to the low number of preocclusive stenosis (90-99% 12 patient). It must be stated that no patients with ipsilateral carotid artery occlusion underwent any revascularization in our institute in concordance with the ESVS guidelines.
  3. . In Table 3, I suggest the authors to present the statistical analysis and the statistically significant difference between the 3 groups.
    • CoW groups were formed as follows: I) compromised CoW contains at least one non-visualized segment; II) non-compromised CoW contains hypoplastic and normal segments. Thus there is a significant difference in the incidence of normal, hypoplastic and non-visualized segments of CoW morphologies between the non-compromised and compromised Cow group, because this is the definition of the grouping. 
      The previously published incidence of non-visualized segments among the anterior communicating artery, anterior cerebral artery, posterior communicating artery, posterior cerebral artery will be added to the discussion section. The number of enrolled patients and the variable occurrence of the different segment variations do not allow further subgroup analysis involving the separation of hypoplastic and normal segments of CoW. This was added to the discussion section. . 

Thank you for your review, we hope our answers here and modifications in the manuscript will provide exact information for you and fulfill the major revision which was asked from you.